# Organic Materials Promote *Rhododendron simsii* Growth and Rhizosphere Soil Properties in a Lead–Zinc Mining Wasteland

**DOI:** 10.3390/plants13060891

**Published:** 2024-03-20

**Authors:** Yunchun Chen, Wei Li, Xinchen Cai, Bo Li, Fangdong Zhan, Yanqun Zu, Yongmei He

**Affiliations:** 1College of Water Conservancy, Yunnan Agricultural University, Kunming 650201, China; ynkmcyc2006@163.com; 2College of Resources and Environment, Yunnan Agricultural University, Kunming 650201, China; ynauweili@126.com (W.L.); zsdscxc@gmail.com (X.C.); ecolibo@foxmail.com (B.L.); zfd97@ynau.edu.cn (F.Z.); zuyanqun@ynau.edu.cn (Y.Z.)

**Keywords:** vegetation reconstruction, *Rhododendron simsii*, organic materials, soil improvement, structural equation model

## Abstract

The mining of metal minerals generates considerable mining wasteland areas, which are characterized by poor soil properties that hinder plant growth. In this study, a field plot experiment was carried out in the mining wasteland of the Lanping lead–zinc mine in Yunnan Province to study the effects of applying three organic materials—biochar (B), organic fertilizer (OF), and sludge (S)—at concentrations of 1% (mass fraction), on promoting the soil of mining wasteland and the growth of two plant varieties (*Huolieniao* and *Yingshanhong*). The results showed that the amount of available nutrients in the surface soil of a mining wasteland could be considerably increased by S and OF compared to the control check (CK). In the rhizosphere soils of two *Rhododendron simsii* varieties, the application of S increased the available phosphorus (P) content by 66.4% to 108.8% and the alkali-hydrolyzed nitrogen (N) content by 61.7% to 295.5%. However, the contents of available cadmium (Cd) and available lead (Pb) were reduced by 17.1% to 32.0% and 14.8% to 19.0%, respectively. Moreover, three organic materials increased the photosynthetic rate and biomass of two *R. simsii* varieties. Specifically, OF and S were found to significantly increase the biomass of *R. simsii*. Organic materials have direct impacts on the increased plant height and biomass of *R. simsii*. Additionally, organic materials indirectly contribute to the growth of *R. simsii* by reducing the content of available Cd and available Pb in rhizosphere soil while increasing the content of available nutrients according to the structural equation model (SEM). Overall, S can stabilize Cd and Pb, increase soil nutrient contents, and promote the growth of *R. simsii* effectively, and has great potential in the vegetation reconstruction of mining wasteland.

## 1. Introduction

China ranks first in the world in terms of Pb ore output (2.4 million tons) and zinc (Zn) ore output (4.5 million tons) [1]. Mineral resources constitute a crucial cornerstone of China’s economic development [2] but at the cost of severe environmental degradation. During mineral exploitation, the outflow of wastewater [3], stacking of waste residue and tailings [4], and leaching contribute to the accumulation of significant amounts of heavy metals in the mining area and surrounding soil. Cd and Pb levels in mining areas were 7.0% and 1.5% greater than normal soil levels, respectively, according to the “National Soil Pollution Survey Bulletin” issued by China in 2014 [5]. The release and movement of heavy metals can result in regional pollution, damage to ecosystems, and threats to human health through contact, respiration, and the food chain, significantly impacting people’s livelihoods and well-being [6,7].

Reclaiming soil contaminated by heavy metals by applying amendments has garnered extensive interest [8]. Utilized as a soil amendment [9], biochar is a byproduct rich in carbon (C) that is produced during the anoxic or oxygen-limited pyrolysis of organic materials such as crop straw [10]. The physicochemical properties of soil can be promoted by biochar, as can its nutrient content and availability [11], microbial community structure and activity, ability to encourage plant growth [12], and ability to reduce heavy metal contamination [13]. The utilization of organic fertilizer is crucial for enhancing plant growth. It serves as a significant approach to improving soil fertility, developing soil chemical properties, maintaining a balanced population of soil microorganisms, optimizing fertilizer efficiency, and contributing to the quality of agricultural produce [14]. The process of sludge decomposition has the potential to enhance the nutrient content and significantly diminish the concentration of heavy metals in the soil of mining-related areas [15]. This process is also widely used in landscaping construction and forested lands [16]. One of the crucial and comparatively easy methods for improving the soil quality of abandoned mining land is the application of a soil conditioner. By modifying the soil pH, altering the form of heavy metals, optimizing the soil structure, and regulating microorganisms, the damage to the soil on abandoned land in mining areas can be reduced, thereby fostering vegetation and reducing heavy metal pollution [17].

The vegetation reconstruction of mine ecological restoration land has gained significant attention from domestic and international scholars in recent years [18]. Planting appropriate vegetation is a widely used biological method for improving soil quality in mining wastelands. However, research is scarce regarding the impact of soil physicochemical property characteristics, pioneer plant development, and the accumulation of Cd and Pb in the mining wasteland of the lead–zinc mine area on the plateau, particularly concerning the combined impacts of various modifiers and pioneer plant cultivation.

*R. simsii* serves as a landscape plant and can be utilized as a pioneer plant for the ecological restoration of mining wastelands, providing aesthetic and environmental benefits [19]. Moreover, *Molinia caerulea* L. is a species with high adaptability to ore tailing landfills that were heavily contaminated with heavy metals around Zn and Pb [20]. In this study, two *rhododendron* varieties (*Huolieniao* and *Yingshanhong*) [21] with high growth adaptability in the mining area were used as experimental plants. The effects of biochar, organic fertilizer, and sludge on improving the soil quality of mining wasteland and the growth of two varieties of *Rhododendron* were evaluated in the mining wasteland of the Lanping lead–zinc mining area in Yunnan Province. This study aimed to investigate the impact of various improved materials on the accumulation of soil nutrients and heavy metals in the mining wasteland of the Lanping lead–zinc mining area. By examining the technical measures of biological improvement and vegetation reconstruction, this study revealed the effects of these measures on soil quality and vegetation in the mining wasteland. The findings of this study can provide valuable guidance for improving soil quality and implementing vegetation reconstruction in mining wastelands.

## 2. Results

### 2.1. Effect of Organic Materials on Soil pH

*Rhododendron* rhizosphere soil was alkaline after being treated with B, OF, and S. The rhizosphere soil of *Yinghanhong* treated with S was weakly alkaline, and the rhizosphere soil of *Huolieniao* treated with B had the highest pH value, at 8.13. The addition of OF and S considerably lowered the soil pH (*p* < 0.05) compared to that of the CK (Table 1). The application of S in this research lowered the pH of the soil because its alkalinity has a buffering and neutralizing effect on the acidity of the soil. It is also possible that the pH of the soil is altered by the decomposition of alkaline substances through organic matter after compost sludge enters the soil [22].

### 2.2. Available Nutrient Contents in Rhizosphere Soil of R. simsii

After the application of organic materials for 60 days, the rhizosphere soils of *Huolieniao* plants treated with B or S exhibited increases in the available P content of 88.0% and 189.9%, respectively (*p* < 0.05), while the rhizosphere soils of *Yingshanhong* exhibited increases in the available P content of 63.5% and 208.8%, respectively (*p* < 0.05). After a 120-day period, the B, OF, and S treatments resulted in 25.3%, 23.4%, and 108.8% increases, respectively, in the amount of available P in the rhizosphere soil of *Huolieniao* (*p* < 0.05). The use of OF and S enhanced the available P content in the rhizosphere soil of *Yingshanhong* by 25.8% and 61.7%, respectively (Figure 1).

The application of organic materials resulted in a significant increase in the alkali-hydrolyzed N content in the rhizosphere soil of the *Huolieniao* and *Yingshanhong* varieties after a period of 60 days. Compared to those in the CK, the OF treatment led to corresponding increases of 111.6% and 186.7%, respectively (*p* < 0.05). After 120 days, the alkali-hydrolyzed N content in the rhizosphere soil of *Huolieniao* was 82.3–92.3% greater in the three treatment groups than in the CK (*p* < 0.05). S had the most significant impact on the alkali-hydrolyzed N content in the rhizosphere soil of *Yingshanhong*, which was 295.5% (Figure 1).

Applying organic materials for 60 and 120 days resulted in a substantial increase in the available potassium (K) content in the rhizosphere soil of the two *R. simsii* species under investigation compared to CK. The application of B had the most significant impact, increasing the value by 3.7–6.6 times (Figure 1). The order of K content increase in the three treatments was as follows: B, S, and OF.

### 2.3. Cd/Pb Content and Speciation in the Rhizosphere Soil of R. simsii

The findings indicated that the use of organic materials for 120 days resulted in a 17.1%, 14.0%, and 21.0% reduction in the available Cd content in the rhizosphere soil of *Huolieniao* under the three treatments of B, OF, and S, respectively, compared to that in the CK (*p* < 0.05). Moreover, the three treatments resulted in a reduction in the available Cd content in the rhizosphere soil of *Yingshanhong*. Notably, the S treatment exhibited the highest available Cd content (32.0%) (Figure 2). The efficacy of S application in diminishing the available Cd concentration in the soil of a lead–zinc mine wasteland is evident.

The levels of available Pb and Cd in the rhizosphere soils of *Huolieniao* and *Yingshanhong* exhibited identical decreasing patterns with the addition of the three organic materials. The most substantial reduction in available Pb content was observed during the S treatment. The available Pb content in the rhizosphere soil of *Huolieniao* and *Yingshanhong* exhibited respective decreases of 22.4% and 19.0% (Figure 2).

Compared to those in the CK, the oxidizable, reducible, and acid-extractable Cd in the rhizosphere soil of *Huolieniao* increased, while the residual Cd in the CK decreased. The most substantial change occurred under S treatment. The oxidizable, reducible, and acid-extractable Cd concentrations increased by 12%, 15%, and 15%, respectively, while the residual Cd concentration decreased by 42%. Similar speciation patterns were found for Pb. With increasing S treatment, the three speciation patterns increased by 2%, 3%, and 12%, whereas the residual fraction decreased by 17% (Figure 3).

With the application of three organic materials, the amount of oxidizable, reducible, and acid-extractable Cd in the rhizosphere soil of *Yingshanhong* increased, while the amount of residual Cd decreased. The modification of S treatment was the most crucial. Compared to those in the CK, the three speciation patterns decreased by 7%, 13%, and 10%, respectively, and the residual Cd increased by 30%. The percentages of acid-extractable Cd and the reducible fraction decreased by 2% and 9%, respectively, with S treatment. The reducible Pb content increased by 9% and constituted the largest percentage, accounting for 47% to 75% of the total (Figure 3).

### 2.4. Effects of Three Organic Materials on the Growth of R. simssi

Similarly, compared with those in the CK, the chlorophyll content in *Yingshanhong* plants increased by 96.8% to 192.9%, the photosynthetic rate increased by 44.9% to 144.1%, and the transpiration rate increased by 34.2% to 66.4%. Once three organic materials were introduced to *Huolieniao plants*, the photosynthetic rate increased by 183.4% to 213.3% (Figure 4). In the presence of B and S, the chlorophyll content increased by 72.6% and 24.6%, respectively. The B and OF treatments resulted in 29.4% and 27.2% increases, respectively, in the transpiration rate and 8.7% to 8.8% increases, in intercellular CO_2_.

Following a 60-day period, the portions of *Yingshanhong* treated with S that were shoots and roots were, respectively, 35.7% and 77.8% higher than those treated with CK (Figure 5). After 120 days, the *R. simsii* that were given organic materials grew better than the CK. To be more precise, the *Huolieniao* variety’s biomass was considerably higher than the CK (*p* < 0.05). Similarly, the biomass of the *R. simsii* treated with OF and S was significantly greater than that of the control treatment (*p* < 0.05).

When three organic materials were applied to *Huolieniao* plants, there was no significant difference between the CK and experimental groups in terms of branch number, uppermost branch number, or root volume. The plant height and canopy height of the B, OF, and S treatments were significantly greater than those of the CK. Specifically, the canopy density of *Huolieniao* was 30.3%, 35.7%, and 22.9% greater than that of the CK under the B, OF, and S treatments, respectively. Similarly, plant height increased by 11.5%, 11.0%, and 9.9% in the three treatment groups compared to the CK (Table 2).

The branch number, uppermost branch number, uppermost leaf number, and root volume of *Yingshanhong* did not differ significantly across the three organic materials and the CK. However, the canopy content significantly increased by 50.8% under the OF treatment compared to the CK. Moreover, the plant height indices for *Yingshanhong* plants under the B and OF treatments were considerably more significant than that of the control group plants by 13.6% and 10.4%, respectively.

### 2.5. The Accumulation of N, P, K, Cd, and Pb in R. simsii

After 120 days of organic material application, the N content in the shoots of *Yingshanhong* plants under the S treatment was significantly greater than that in the CK. The N content was 151.6% and 37.1% greater in the B and S treatments, respectively, than in the CK. Similarly, for *Huolieniao*, the N content in the shoot was 36.2% to 42.6% greater in the OF treatment than in the CK (Figure 6). These observations suggest that S treatment had a more significant impact on N content than OF treatment, indicating the effectiveness of utilizing S as an organic material for promoting N content in *R. simsii*.

After 120 days of organic material application, the P levels in the shoots and roots of *Huolieniao* were significantly greater than those in the control group plants, at 74.1~116.6% and 25.4~113.0% greater, respectively, than those at 60 days (Figure 6). After a 120-day period during which three organic materials were applied, the shoot of *Yingshanhong* exhibited a P content significantly greater than that of the CK, ranging from 164.9% to 270.2%.

After 60 days of applying organic materials, there was no notable difference in K levels between the *Huolieniao* treatment and the CK. After 120 days, the concentration of K in the shoots of *Huolieniao* plants treated with organic materials was significantly greater than that in the control group plants, with an increase ranging from 17.2% to 44.0% (Figure 6). There was no discernible difference in K concentration between the shoots of *Yingshanhong* and those of the CK following the addition of organic materials. Adding B and S significantly increased the K content in the roots of *Yingshanhong* compared to that in the CK (*p* < 0.05).

The Cd content in the shoots of *Huolieniao* decreased after 120 days of organic material application. The application of OF and S resulted in decreases of 45.2% and 36.2%, respectively, in the Cd level in the shoots of *Huolieniao*. The Pb content in the shoot remained stable, but the Pb level in the root decreased by 27.3% to 52.7% (Figure 7). The concentration of Cd in the roots of *Yingshanhong* significantly decreased (*p* < 0.05), while the levels of Pb in both the shoots and roots decreased, with decreases ranging from 34.1% to 49.0% and 12.4% to 53.1%, respectively. The utilization of organic materials has been observed to significantly decrease the levels of Cd and Pb in *R. simsii*.

### 2.6. Correlations between Plant Growth and Soil Available Nutrients and Different Fractions of Heavy Metals

A significant positive correlation (*p* < 0.01) was observed between soil pH and soil available K and oxidizable Pb. There was a statistically significant positive correlation (*p* < 0.001) between the soil available P and shoot biomass in *Huolieniao*. A significant positive correlation was observed between soil alkali-hydrolyzed N and the shoot biomass, canopy, and plant height (*p* < 0.001) of *Huolieniao*. A significant positive correlation was also found between N and root biomass (*p* < 0.05). There was a statistically significant negative correlation (*p* < 0.01) between the shoot biomass of *Huolieniao* and the residual Cd and Pb (Figure 8). The results suggest that using organic materials can passivate the available Cd and Pb in the rhizosphere soil of *R. simsii* and potentially stimulate the growth of *R. simsii*.

The shoot biomass of *R. simsii* was significantly positively correlated with the soil available P, available K, and residual Pb (*p* < 0.05). A significant positive correlation (*p* < 0.05) was observed between soil alkali-hydrolyzed N and the uppermost leaf number, canopy, and shoot biomass of *R. simsii*. There was a significant negative correlation (*p* < 0.01) between the available Pb and oxidizable Cd and the plant height of *R. simsii*. Similarly, the biomass of *R. simsii* was negatively correlated with the amount of reducible Cd (*p* < 0.05) (Figure 9). Evidently, applying organic materials can enhance nutrient availability in the soil, consequently stimulating the development of *R. simsii*.

The structural equation model, which was developed utilizing three organic materials, accounted for 90.0% of the biomass and 93.0% of the plant height of *Huolieniao*. The overall assessment of soil nutrient availability ranged from 88.8% to 100%. Organic materials positively affected the available K, soil alkali-hydrolyzed N, plant height, and biomass of *Huolieniao* plants, with respective total effects of 0.96, 0.35, 0.75, and 0.62 (Figure 10). Additionally, the plant height of *Huolieniao* was positively impacted by the presence of P, the values of which were 0.53, −0.03, and −0.87, respectively, which were the values of available Cd and P that were adversely affected by soil pH.

The height and biomass of *Yingshanhong* were explained by a structural equation model composed of three organic materials, accounting for 80.0% and 91.0% of the variance, respectively. Alkaline N and available P were interpreted as totaling 86.0% and 84.0%, respectively. The height and biomass of *Yingshanhong* were all positively impacted by organic materials, as demonstrated by the model; these impacts added up to 0.16, 0.35, 0.60, and 0.69, respectively (Figure 10). *Yingshanhong*’s biomass, which was 0.57, was also positively impacted by available P. Like in *Huolieniao*, available P and Cd were negatively impacted by soil pH, with respective values of −0.88 and −0.06. Organic materials hurt the amount of heavy metals in the soil. *R. simsii* plant height and biomass can be directly influenced by organic materials, and they can also have an indirect effect by altering the pH, available Cd, available Pb, and available nutrient content of the soil.

## 3. Discussion

### 3.1. Effects of Three Organic Materials on the Promotion of Soil Available Nutrients

The lead–zinc mining area is characterized by depleted soil, elevated heavy metal concentrations that surpass acceptable limits, and severe contamination [23]. Increasing soil fertility in lead–zinc mining regions has become an international and domestic research concern in recent years. Organic materials increase soil fertility for the following three reasons according to Luo’ findings [24]: 1. The soil contains a substantial amount of organic matter, along with elements such as N, P, K, and magnesium, which effectively counteract soil alkalinity and avert salinization and solidification. 2. Prolonged fertility effectively discourages rainwater erosion and loss. 3. Organic matter facilitates nutrient release and ferments the soil’s oxygen content and porosity via fermentation.

P and K availability in the soil were significantly enhanced by the addition of biochar in this study, which is consistent with the findings of Zhu [25]. This phenomenon could be attributed to the fact that biochar ash contains water-soluble mineral elements that can increase both the total nutrient content of cultivated soil and the quantity of nutrients accessible to crops [26]. Additionally, the acidic functional groups and metal oxide hydroxylation surface of biochar enable it to adsorb and retain mineral cations; this property enhances the cation exchange capacity (CEC) value of cultivated soil and mitigates nutrient leaching loss [27,28]. When the CEC content is high in soil, the effect of fixing heavy metals will generally be better. The soil cultivation experiments carried out by Gondek [29] showed that the addition of biochar from poultry manure increased the CEC of soil, which was one of the reasons for the enhanced passivation effect of heavy metals in soil. Moreover, by continuously generating oxygen-containing functional groups on the surface of biochar, the continuous slow oxidation reaction of biochar in the soil can increase the soil’s CEC, regulating the sustainable productivity of cultivated soil [30].

Organic fertilizer is produced by utilizing biological substances, including animal and plant waste. As a result, plant manure contains the essential elements (N, P, and K) needed for crop development and the trace elements of calcium and magnesium. Moreover, the concentrations of these elements are significantly greater than those found in soil [31]. Conversely, the utilization of organic fertilizer might introduce a substantial amount of organic material [32]. Organic matter, a crucial component of the soil solid phase, is vital in providing nutrients over an extended period and preventing nutrient depletion. The use of organic fertilizer in this investigation contributed to a substantial increase in the concentration of alkali-hydrolyzed N, which was consistent with the findings of Wu [33].

The use of sludge has a noteworthy enhancing impact on the levels of soil available P, available K, and alkali-hydrolyzed N. Among these factors, the impact on the concentration of available P is particularly pronounced. A research by Kołodziej revealed that the application of 40 Mg ha^−1^ DM of sludge resulted in the highest yield of biomass of *Bamse* [34]. Sonia [35] posited that sludge can promote soil fertility and stimulate plant development, a finding that aligns with the outcomes of our investigation. Plants can directly absorb a significant quantity of N and P present in sludge, and sludge contains high concentrations of organic N and P. Upon application to the soil, organic N and P undergo a progressive process of mineralization, facilitated by the reduced carbon-to-N ratio, which in turn enhances the mineralization of the sludge. Simultaneously, organic matter undergoes a partial transformation into soil humus, stabilizing and gradually releasing nutrients. This process has a beneficial fertilizing impact on the soil [36].

### 3.2. Effects of Organic Materials on the Bioavailability of Cd and Pb in Soil

The primary function of amendments is to decrease the concentration of available heavy metals in soils contaminated with heavy metals. Three types of organic materials have been found to effectively decrease the levels of available Cd and Pb in soil while also exhibiting a passivating impact on soil heavy metals, and these findings align with the studies conducted by Xie [37] and Mehmood [38].

Huang [39] observed that mineral additions significantly altered the distribution of Cd, Pb, Cu, and Zn species in soil, transforming them from readily exchangeable forms to those with poor bioavailability. Due to its extensive surface area and high concentration of oxygen-containing functional groups, such as carboxyl and phenol hydroxyl groups, biochar can effectively bind heavy metals to its surface through chelation, and the bioavailability of heavy metals is reduced by this process [40]. The consumption of 1% biochar (mass fraction) resulted in a 72.0% reduction in the amount of soil Cd under accessible conditions according to Wu’s findings [41]. Chen [42] demonstrated that including 3% of biochar (mass fraction) resulted in a substantial decrease in the soil Pb concentration. The use of 1% biochar (mass fraction) in this study resulted in reductions of 21.0% and 23.8% in the concentrations of Cd and Pb, respectively, in the rhizosphere soil of *Yingshanhong*. Netherway [43] found that pyromorphite and lead phosphate were formed due to the precipitation reaction between Pb and phosphate on biochar, and Pb could be effectively immobilized by utilizing poultry manure biochar in Pb-contaminated soil. These findings demonstrated that biochar has specific potential for use in addressing soil Cd contamination. Organic fertilizer contains a high amount of organic matter, which is abundant in functional groups and aromatic structures. These heavy metals can absorb and chelate heavy metals present in the soil. Furthermore, organic fertilizer can not only suppress the activity of heavy metals by augmenting the cation exchange capacity of soil, but also boost the diversity and activity of microorganisms in soil, thereby exerting a substantial impact on the remediation of heavy metals [44]. The primary function of sludge is to diminish the efficacy of heavy metals. For instance, the interaction between activated sludge and soil might generate more adsorption sites, strengthen the ability of soil to adsorb Cd and antimony, and decrease their bioavailability [45].

### 3.3. Effects of Three Organic Materials on Plant Growth Characteristics

Deng [46] reported that utilizing organic fertilizer led to a substantial increase in the basal diameter and root activity of *Vaccinum corymbosum* seedlings. Additionally, these changes resulted in increased total branch length, total leaf area, and biomass. In their study, Deng [47] noted that the combined application of rice husk biochar and compound fertilizer had a significant impact on the growth of *Phoebe zhennan*. Specifically, compared with single-application compound fertilizer, this treatment led to increased plant height, leaf area, as well as enhanced N absorption by the leaves. The use of sawdust biochar in conjunction with compound fertilizer substantially increased both the stem biomass and leaf area of *Phoebe zhennan*. Furthermore, this combination facilitates the enhanced uptake of K by plant roots. Applying OF significantly improved the plant height, crown width, and root length of the two varieties of *Rhododendron* in this study. Additionally, the photosynthetic rate increased by 44.9~213.3%, the transpiration rate increased by 27.2~34.2%, and biomass increased by 55.9~82.6%. *Rhododendron* growth can be efficiently promoted by applying organic fertilizer.

Rita [48] concluded that the combination of biochar and organic fertilizer had a significant impact on enhancing soil fertility and boosting *sugar beet* production. Zhou [49] demonstrated that the utilization of biochar resulted in a decrease in soil pH, an increase in soil available P and available K content, and increases in the activities of α-glucosidase, N-β-glucosaminidase, phosphatase, and sulfatase enzymes. Additionally, it significantly enhanced the photosynthetic traits, biomass, root structure, and N and P uptake of *Rhododendron*. These outcomes align with the findings of the present study.

The plant height and canopy height of the B, OF, and S treatments were significantly higher compared to the CK treatment in this study. In particular, *Huolieniao*’s canopy density under the B, OF, and S treatments was 30.3%, 35.7%, and 22.9% higher than that of the CK, respectively. In the same way, the three treatment groups’ plant height rose by 11.5%, 11.0%, and 9.9% compared to the CK. The plant height indices for *Yingshanhong* plants under the B and OF treatments were significantly greater than those of the control group plants by 13.6% and 10.4%, respectively. However, the canopy content increased significantly by 50.8% under the OF treatment compared to the CK. These findings showed that the application of S can effectively improve the available nutrients in the rhizosphere soil of *Rhododendron*, increase the biomass of *Rhododendron*, and reduce the content of the heavy metals Cd and Pb in *Rhododendron*. The structural equation model (SEM) showed that organic materials negatively affected the available heavy metals in soil. Organic materials can directly affect the plant height and biomass of *Rhododendron*. They can also indirectly affect their plant height and biomass by changing the pH, available Cd, available Pb, and available nutrient content in the soil.

This investigation revealed that the N, P, and K contents of the two *R. simssi* varieties increased in response to the application of the three organic materials. Conversely, the Pb and Cd contents of *Yingshanhong* decreased due to S treatment. This finding demonstrated how S application could efficiently increase the biomass of *R. simssi*, reduce the concentrations of the heavy metals Pb and Cd, and increase the availability of nutrients in the rhizosphere soil of *R. simssi*. Structural equation modeling (SEM) revealed that organic materials negatively affected the available heavy metals in the soil. The height and biomass of *R. simssi* plants can be directly influenced by organic materials and be indirectly affected by alterations in soil pH, available Cd, available Pb, and available nutrient content.

## 4. Materials and Methods

### 4.1. Study Area

The experiment was conducted at the Lailong Village (26°48′28′′ N, 99°28′38′′ E), Jinding Town, Lanping County, Nujiang Prefecture, Yunnan Province, China. The site inclination angle is 36°, and the altitude is 2775 m. The site is located northwest of Yunnan Province and has a low-latitude mountain monsoon climate, with a mean annual temperature of 13.7 °C, a maximum mean monthly temperature of 25.5 °C (July), and a minimum mean monthly temperature of 3.4 °C (January). The frost-free period is 190 days, and the mean rainfall is 158 days. The rainy season begins in late May and ends in mid-October, and the mean annual precipitation is 1002 mm.

The surface soil (0–20 cm) in the mining wasteland was collected and transported back to the laboratory. After the removal of plant debris, the soil was air-dried and sieved through a 100 mm mesh screen for use. The pH of the collected soil was 7.86, the organic matter content was 7.96 g/kg, the available P content was 10.6 mg/kg, the available K content was 41.3 mg/kg, the alkali-hydrolyzed N content was 19.2 mg/kg, the Cd content was 226 mg/kg, and the Pb content was 16234 mg/kg.

### 4.2. Experimental Design

Two *R. simsii* varieties (*Huolieniao* and *Yingshanhong*, Jinhua Yonggen Rhododendron Cultivation co., Ltd., Jinhua, China) were selected, and 24 experimental plots were established in July 2021 with a 2.0 × 1.5 m long planting grid. A 0.5 m wide isolation belt was set up to eliminate interference between the cells. In this experiment, 4 treatments were used: (1) CK, no treatment; (2) biochar; (3) organic fertilizer (Yunnan Huaning Xichao Agricultural Technology Co., Ltd., Huaning, China); and (4) sludge (sewage treatment plant, Jinning District, Kunming City) (Table 3). Each treatment was repeated 3 times, and an aliquot of raw 1% (mass fraction) organic material (~6.75 kg) was thoroughly mixed with the soil. A total of 20 plants (4 × 5) of the same species of *R. simsii* were planted in each plot. The biochar was prepared by crushing the branches of the tested rubber trees under anaerobic conditions at 400~500 °C. All plots were randomly distributed, and a 0.5 m wide isolation belt was used to compensate for differences in light and temperature in the field.

### 4.3. Sampling and Measurements of Soil Properties

The surface soil (0–20 cm) in each plot was collected by a soil auger according to the diagonal sampling method of [50] and subsequently transported to the laboratory. After removing plant debris, the soil was air-dried and sieved through 100- and 20-mesh nylon sieves to determine the basic properties and heavy metal content, respectively.

The height of the *R. simsii* plants was measured for each plot using a stainless steel scale with a precision of 1 cm. After the biomass sample was immobilized and dried to a constant weight, its mass was determined using an electronic scale with a precision of 0.001 g. The method utilized to determine the total chlorophyll content was ethanol extraction with an ultraviolet–visible spectrophotometer. A 25 mL volumetric flask was used to maintain a constant volume, ethanol and calcium carbonate were combined with 0.200 g of sample, and the leaves were ground until completely ground. To compare the colors at 665 and 649 nm, an ultraviolet–visible spectrophotometer was used (Thermo Fisher Scientific Shier Technology Co., Ltd., Waltham, MA, USA). The leaf transpiration rate and intercellular CO_2_ concentration were recorded from 8:00 am to 11:00 am in clear weather with Li-6400 portable photosynthetic equipment (LI-COR Biosciences, Lincoln, NE, USA) under natural light conditions. N in the plants was measured by the acid digestion-modified Kjeldahl method; P in the plants was measured by the acid digestion-molybdenum antimony resistance colorimetric method, and K in the plants was measured by the acid digestion-flame atomic absorption method [51].

The pH was measured using a 1:2.5 soil/solution ratio in distilled water [50]. The organic matter content was measured using the K dichromate volumetric method [50]. The alkaline N content was measured by using the alkali-hydrolyzed N diffusion method [50]. The available P was quantified using the molybdenum antimony colorimetric method [50]. The available K was determined using CH_3_COONH_4_ extraction flame spectrophotometry [50] (Shanghai Instrument Electroanalytical Instrument Co., Ltd., Shanghai, China). Determination of available Cd and Pb in the soil [52]: 0.5 g of 20-mesh sieved soil was weighed, 25 mL of DTPA was added, the mixture was oscillated for 2 h, and subsequently filtered and analyzed via a flame atomic absorption spectrometer (Thermo Fisher Scientific Shier Technology Co., Ltd., Waltham, MA, USA).

The air-dried soil samples screened at 0.15 mm were digested with aqua regia-perchloric acid and concentrated nitric acid at high temperature, and the total Pb content in the soil was determined by flame atomic absorption spectrometry (Thermo Fisher Scientific Shier Technology Co., Ltd., Waltham, MA, USA). The method developed by Tessier [53] was used to fractionate the Cd and Pb content into carbonates (acid-extractable), Fe and Mn oxides (reducible), organic matter (oxidizable), and residual fractions. Metals bound to carbonates—the form of a exchangeable residue was extracted with 8 mL of 1.0 mol/L CH_3_ COONa, pH 5, by shaking constantly for 5 h at 25 °C. Metals bound to Fe and Mn oxides—carbonate residue was extracted for 5 h 20 mL of 0.04 mol/L NH_2_OH.HCl dissolved in 25% CH_3_COOH (*v*/*v*) at 96 °C, stirring from time to time. Metals bound to organic matter—residue form of Fe-Mn oxides was extracted 3 mL 0.02 mol/L HNO_3_ + 5 mL of 30% H_2_O_2_ (*v*/*v*) for 2 h at 85 °C, stirring occasionally, and then 3 mL of 30% H_2_O_2_ was added at 85 °C while stirring from time to time. Finally, the solution was shaken with 5 mL of 3.2 mol/L CH_3_COONH_4_ in 20% HNO_3_ (*v*/*v*) for 0.5 h at 25 °C. Metal residue bound to residual fraction—the remains of organic form was hot-decomposed with 10 mL of 40% HF and 2 mL of 70% HClO_4_; 1 mL HClO_4_ was added after drying, and heated until white fumes appeared. The residue was dissolved in HCl (1:1) and diluted to 100 mL. The extracts were analyzed for Cd and Pb using an acetylene-flame atomic absorption spectrometer (Thermo Fisher Scientific Shier Technology Co., Ltd., Waltham, MA, USA).

### 4.4. Statistical Analysis

The differences in soil properties, Pb concentration and proportion of Pb fractions, and Cd concentration and proportion of Cd fractions among the different treatments were compared using two-way analysis of variance (ANOVA) at a significance level of 0.05. Pearson correlation analysis was performed to explore the relationships of the soil properties with the plants. Two-way ANOVA and Pearson correlation analysis were performed with IBM SPSS Statistics 22.0 software (SPSS, Inc., Chicago, IL, USA). Amos 26 (SPSS, Inc., Chicago, IL, USA) was used to construct the structural equation model. Origin 2021 (OriginLab Corporation, Northampton, MA, USA) was used for drawing the manuscript.

## 5. Conclusions

(1) The introduction of OF and S had a substantial impact on the levels of available P, alkali-hydrolyzed N, and available K in the rhizosphere soil of *R. simsii* (*p* < 0.05). In particular, the application of S significantly improved the soil’s condition and had a major effect on the alkali-hydrolyzed N content in rhizosphere soil of *Yingshanhong*, of which was 295.5%.

(2) The utilization of these three organic substances significantly promoted the development of *R. simssi*. Once the three organic materials were introduced to *Huolieniao*, the photosynthetic rate increased by 183.4~ 213.3% and the biomass variety was significantly greater than that of the CK (*p* < 0.05).

(3) Additionally, the three organic substances significantly decreased the levels of accessible Cd and accessible Pb in the rhizosphere soil of *R. simsii*, and integrating S strongly inhibited the availability of Cd and Pb in the soil (*p* < 0.05). This resulted in a 17.1% and 32.0% reduction in the available Cd in the rhizosphere soils of *Huolieniao* and *Yingshanhong* and a 14.8% and 19.0% decrease in the available Pb, respectively. The application of S resulted in increases of 84.3% and 50.2% in the total biomass of *Huolieniao* and *Yingshanhong*, respectively. In conclusion, the utilization of S can effectively increase soil nutrient availability in the Lanping lead–zinc mining area and can also mitigate the soil contamination caused by heavy metals such as Cd and Pb.

## Figures and Tables

**Figure 1 plants-13-00891-f001:**
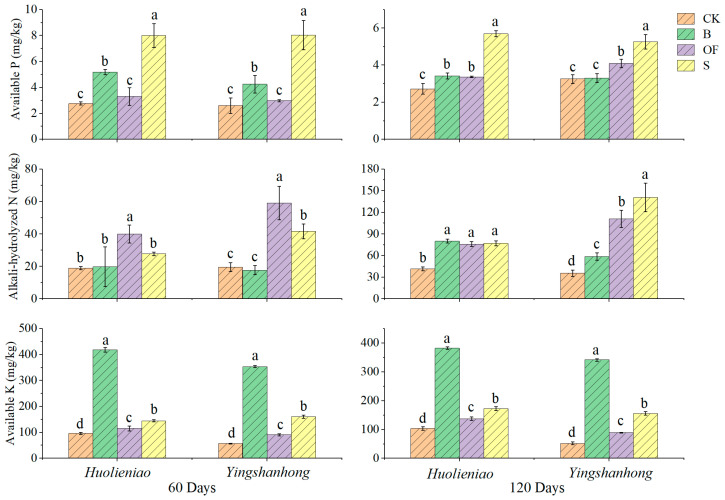
Effects of 3 organic materials on the available nutrients in rhizosphere soil of *R. simsii*. Different lowercase letters indicate significant differences among 3 organic material treatments (Duncan’s comparisons, *p * <  0.05).

**Figure 2 plants-13-00891-f002:**
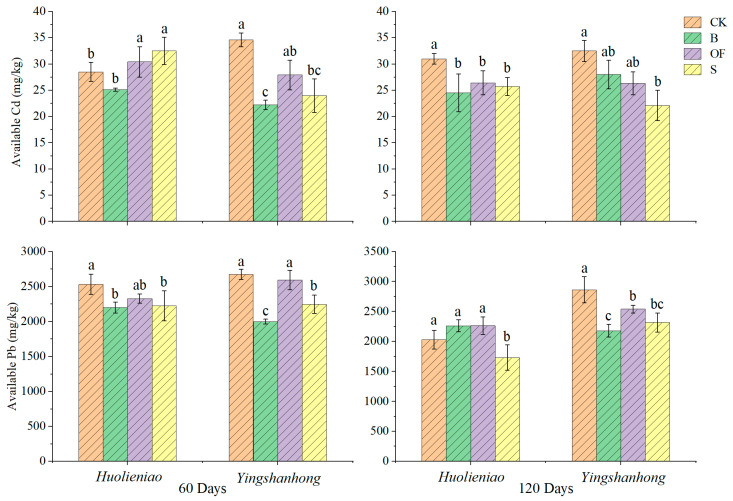
Effects of 3 organic materials on the available Pb and Cd contents in rhizosphere soil of *R. simsii*. Different lowercase letters indicate significant differences among 3 organic material treatments (Duncan’s comparisons, *p * <  0.05).

**Figure 3 plants-13-00891-f003:**
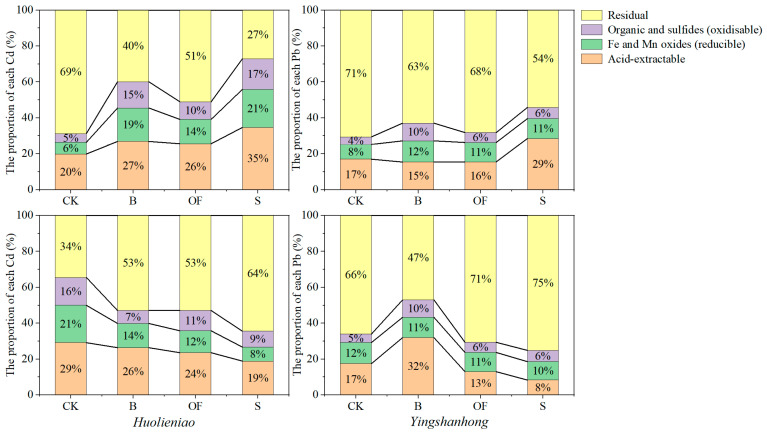
Distribution of different fractions of Cd and Pb in rhizosphere soil of *R. simssi* under 3 organic materials.

**Figure 4 plants-13-00891-f004:**
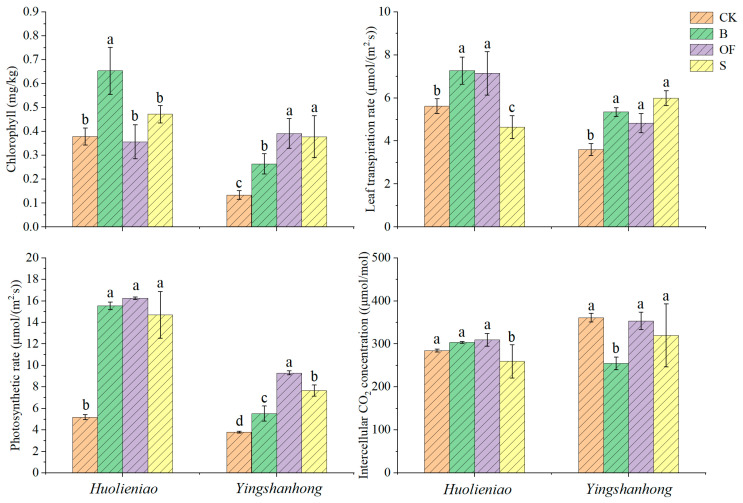
Effects of 3 organic materials on photosynthesis parameters of *R.simsii*. Different lowercase letters indicate significant differences among 3 organic material treatments (Duncan’s comparisons, *p * <  0.05).

**Figure 5 plants-13-00891-f005:**
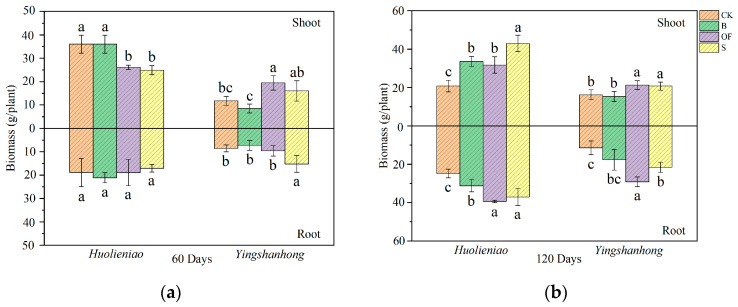
Effects of 3 organic materials on biomass of *R. simsii*: (**a**) 60 days, (**b**) 120 days. Different lowercase letters indicate significant differences among 3 organic material treatments (Duncan’s comparisons, *p * <  0.05).

**Figure 6 plants-13-00891-f006:**
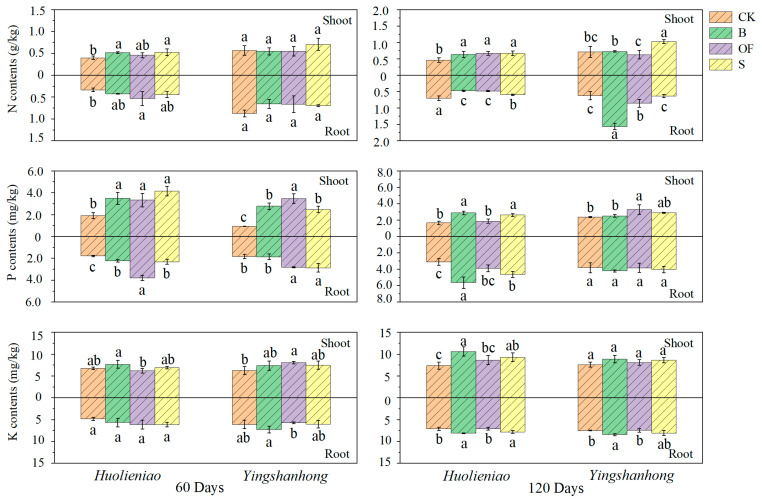
Effects of 3 organic materials on N, P, and K of *R. simsii*. Different lowercase letters indicate significant differences among 3 organic material treatments (Duncan’s comparisons, *p * <  0.05).

**Figure 7 plants-13-00891-f007:**
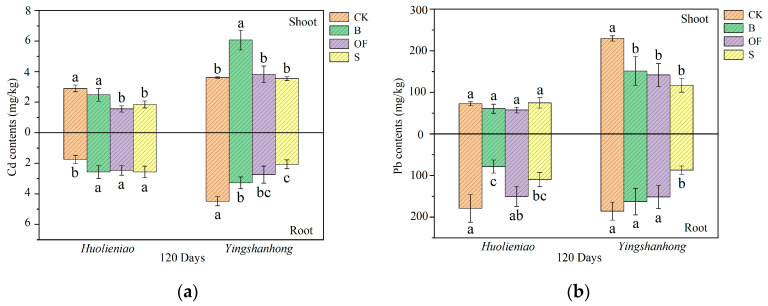
Effects of 3 organic materials on the Cd contents in *R. simsii* (**a**) and Pb contents in *R. simsii* (**b**). Different lowercase letters indicate significant differences among 3 organic material treatments (Duncan’s comparisons, *p * <  0.05).

**Figure 8 plants-13-00891-f008:**
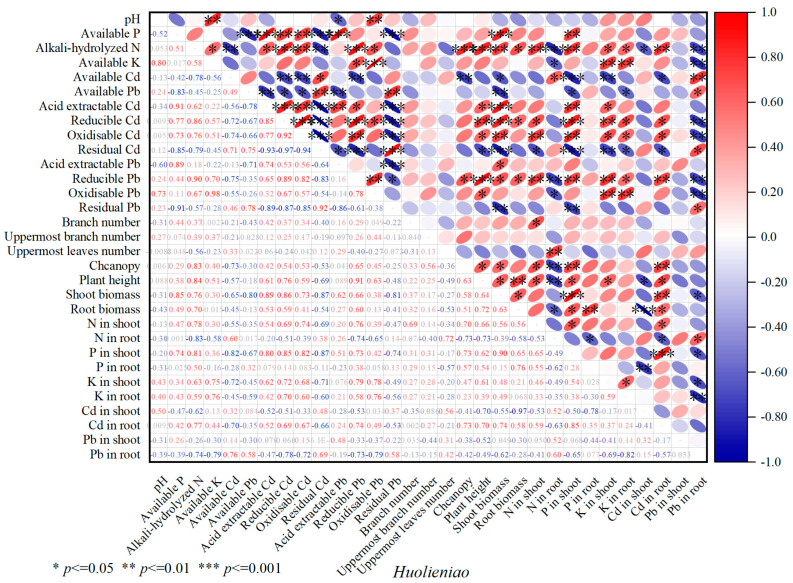
Correlation analysis of soil available nutrition and differentiation morphological heavy metals with plant growth of *Huolieniao* under 3 organic material treatments.

**Figure 9 plants-13-00891-f009:**
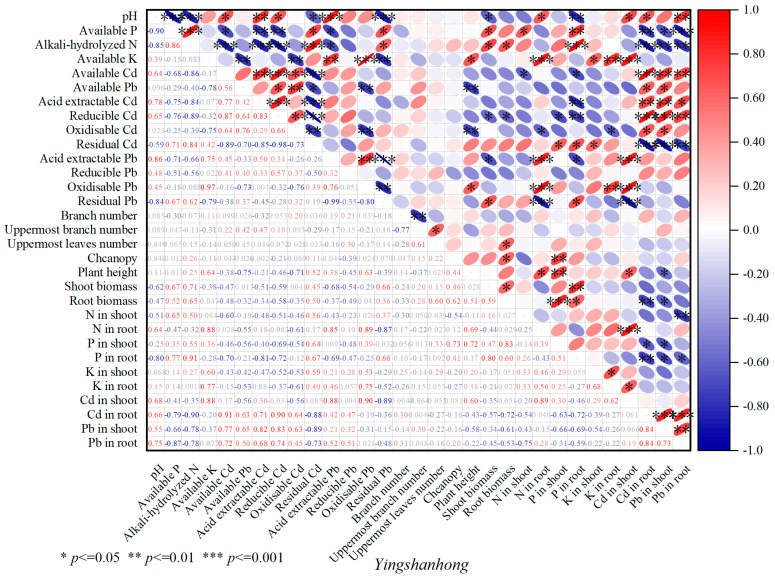
Correlation analysis of soil available nutrition and differentiation morphological heavy metals with plant growth of *Yingshanhong* under 3 organic material treatments.

**Figure 10 plants-13-00891-f010:**
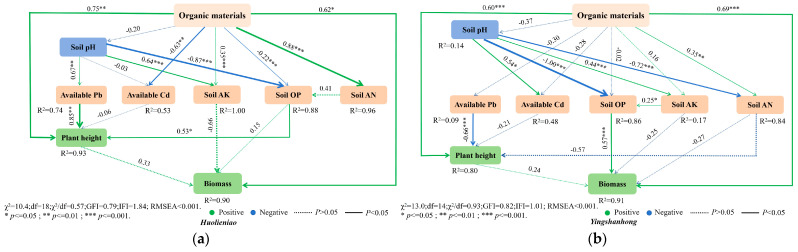
Main pathways of the effects of structural equation model with organic materials on soil available nutrients and growth of 2 *R. simssi*: (**a**) *Huolieniao*; (**b**) *Yingshanhong*.

**Table 1 plants-13-00891-t001:** Effects of 3 organic materials on rhizosphere soil pH of *R. simsii*.

Treatments	*Huolieniao*	*Yingshanhong*
CK	7.80 ± 0.04 b	7.71 ± 0.03 a
B	8.13 ± 0.04 a	7.79 ± 0.03 a
OF	7.67 ± 0.02 c	7.57 ± 0.04 b
S	7.57 ± 0.04 d	7.42 ± 0.05 c

Values followed by different letters (e.g., a, b, and c) in the same row are significantly different among treatments with *p* < 0.05 (Duncan’s test).

**Table 2 plants-13-00891-t002:** Morphological parameters of *R. simsii*.

Plant	Treatments	Plant Height/cm	Chcanopy/cm	Branch Number	Uppermost Branch Number	Uppermost Leaves Number	Root Length/cm	Root Volume/cm^3^
*Huolieniao*	CK	76.5 ± 1.1 b	38.9 ± 4.0 b	6.3 ± 1.0 a	5.2 ± 0.9 a	41.5 ± 15. 1 a	8.0 ± 0.26 b	42.79 ± 5.5 a
	B	85.3 ± 2.3 a	50.7 ± 7.7 a	6.7 ± 0.4 a	6.1 ± 0.7 a	30 ± 1.7 ab	9.44 ± 0.62 a	46.17 ± 6.3 a
	OF	84.9 ± 2.4 a	52.8 ± 3.6 a	7.1 ± 0.4 a	5.8 ± 1.2 a	24.3 ± 2.6 b	8.9 ± 0.35 ab	51.38 ± 8.8 a
	S	84.1 ± 3.0 a	47.8 ± 1.5 a	7.3 ± 0.9 a	5.6 ± 0.4 a	35.6 ± 4.8 ab	9.1 ± 1.0 ab	47.68 ± 7.6 a
*Yingshanhong*	CK	55.7 ± 1. 6 c	32.9 ± 1.9 b	7.3 ± 1.5 a	4.2 ± 0.8 a	5.5 ± 0.2 a	7.3 ± 0.5 b	34. 1 ± 4.9 a
	B	63.3 ± 2.3 a	37.3 ± 2.3 b	7.7 ± 2.0 a	3.3 ± 1.1 a	5.9 ± 0.6 a	8.9 ± 0.7 a	37.8 ± 5.7 a
	OF	61.5 ± 2.1 ab	49.6 ± 7.6 a	7.3 ± 1.5 a	4.4 ± 2.2 a	9.1 ± 4.5 a	8.8 ± 0.9 a	38.2 ± 1.0 a
	S	59.2 ± 1.9 bc	33.4 ± 3.5 b	7.0 ± 0.9 a	3.4 ± 1.5 a	5.5 ± 3.1 a	9.0 ± 0.4 a	39.5 ± 4.8 a

Values followed by different letters (e.g., a, b, and c) in the same row are significantly different among treatments with *p* < 0.05 (Duncan’s test).

**Table 3 plants-13-00891-t003:** Basic chemical properties of the 3 organic materials.

Organic Materials	pH	Organic Matter (g/kg)	Available P (mg/kg)	Available K (mg/kg)	Alkeline N (mg/kg)	Cd (mg/kg)	Pb (mg/kg)
B	8.87	186.7	377.2	357.1	23.7	0.22	5.17
OF	7.14	463.7	49.3	106.5	127.4	0.87	39.0
S	7.43	211.1	245.5	160.5	177.6	1.12	47.6

## Data Availability

Data are contained within the article.

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
