# Peer review of "Organic Materials Promote Rhododendron simsii Growth and Rhizosphere Soil Properties in a Lead–Zinc Mining Wasteland"

_plants, 2024, doi:10.3390/plants13060891_

Round 1

Reviewer 1 Report

Comments and Suggestions for Authors

Title of manuscript to accept

Abstract and key words – correct

What is sludge – sewage sludge from Treatment ….

Introduction

Please mention about others tailings

for instance

Pietrzykowski M., Antonkiewicz J., Gruba P., Pająk M. 2018. Content of Zn, Cd and Pb in purple moor-grass in soils heavily contaminated with heavy metals around a zinc and lead ore tailing landfill. Open Chemistry, 16, 1143-1152. DOI: https://doi.org/10.1515/chem-2018-0129

In discussion

Linie 313

Please add paper to cited

for instance

Kołodziej B., Stachyra M., Antonkiewicz J., Bielińska E., Wiśniewski J. 2016. The effect of harvest frequency on yielding and quality of energy raw material of reed canary grass grown on municipal sewage sludge. Biomass and Bioenergy, 85, 363-370. DOI: http://dx.doi.org/10.1016/j.biombioe.2015.12.025

Material and methods

Were the Certified reference materials (CRM) used in chemical analysis? What was standard reference material?   

Were a mass balance do for metals ? 

Please see below, for instance:

Assessment of heavy metals content tested by the sequential method of Tessier et al. [1979] was verified by the mass balance proposed in the works …… For the n-stage sequential extraction, the sum of the metal fractions in the form of chemical fractions is theoretically equal to the total metal content in the tested sample. The mass balance of heavy metals was performed by comparing the total metal content in the samples by single-stage digestion of the sample with the sum of heavy metals in individual fractions analyzed by sequential extraction. The degree of compliance of the balance of heavy metals was over 95%.

For instance

Antonkiewicz J., Pełka R. 2014. Fractions of heavy metals in soil after the application of municipal sewage sludge, peat, and furnace ash. Soil Science Annual, 65, 3, 118-125. DOI: 10.1515/ssa-2015-0003

Reviewer 2 Report

Comments and Suggestions for Authors

The present study of the article entitled “Organic materials promote Rhododendron simsii growth and rhizosphere soil properties in a lead-zinc mining wasteland” has an interesting area of study. The manuscript required modifications to further enhance the scientific rigor and impact of the study.

The manuscript requires a significant improvement in grammar.

1. In the abstract Abbreviations should be defined in first mention, please revise this issue in the whole ms! The authors should follow abbreviation rule – first define a term, and then use the abbreviated form.
2. Why did the author select biochar (B), organic fertilizer (OF) and sludge (S), at concentrations of 1% (mass fraction).
3. The introduction requires serious revision as there are many grammatical and verbal mistakes.

4. The object and hypothesis of the current study is not clear please clarify more.

5. Figures resolution should be raised. Please improve the quality of the figures. 

6. The interplay between different parameters in response to different organic materials is intricate and interconnected. Discussing potential cross-regulation would provide a more nuanced understanding of the changes observed in this study.

7.  Further elaborate on the mechanistic link between the observed changes in physiological parameters and the improved plant characteristics in the discussion section is strongly suggested. How do the changes in physiological parameters contribute to the observed enhancements in plant growth? Provide hypotheses on the underlying physiological processes.

8. Some papers should be added to enhance discussion part.

9. The conclusion section must be rewritten. Authors should include specific results of their research, which extend the current state of knowledge. Moreover, Suggest possible directions for future research.

10. References: need to be cross-checked.

11. English should be polished. 

Comments on the Quality of English Language

The manuscript requires a significant improvement in grammar.

Reviewer 3 Report

Comments and Suggestions for Authors

In the manuscript "Organic materials promote Rhododendron simsii growth and rhizosphere soil properties in a lead-zinc mining wasteland", the authors used R. simsii as a pioneer plant for the ecological restoration of mining wasteland, providing both aesthetic and environmental benefits. The authors used two Rhododendron varieties (e.g., Huolieniao and Yingshanhong) with high growth adaptability in the mining area for this study. The authors examined the effects of biochar, organic fertilizer, and sludge on improving soil quality of mining wasteland, as well as on the growth of two varieties of Rhododendron in the mining wasteland of the Lanping lead-zinc mining area in Yunnan Province. They studied the impact of biochar, organic fertilizer, and sludge on the accumulation of soil nutrients and heavy metals in the mining wasteland of the Lanping lead-zinc mining area. As the authors concluded, the findings can provide valuable guidance for improving soil quality and implementing vegetation reconstruction in mining wastelands.

This study is interesting, however, revisions are needed as follows:

-          Lines 16-17: Use the abbreviation only for sludge and organic fertilizer, but use the full name of the abbreviation CK for the first time.

-          Throughout the manuscript: Please write numbers less than 10 verbally.

-          Throughout the manuscript: Please write in full the word you want to abbreviate for the first time, with the abbreviation next to it in parentheses, and after that use the abbreviation only.

-          Line 52: “properties characteristics” Choose one of the two words, leave it, then cancel the other.

-          Line 66: The same comment of line 52.

-          Figures must be highly resolution.

-          Line 443: “24 experimental plots were established”, while in line 445 “4 treatments were used”, and in Line 448 “Each treatment was repeated 3 times”, therefore, the experimental plots were 12 not 24, isn’t it?

-          Lines 446-448: It is not clear how the treatments were carried out, and the section (Experimental design) in general needs more details. For example, how much biochar, organic fertilizer, and sludge were added? On what basis were these quantities applied to the soil chosen? and so on.

-          The results showed that the sludge lowered the soil pH more than the organic fertilizer, even though the organic fertilizer had a lower pH than sludge. Explain.

In general: There are many grammatical errors throughout the manuscript, please correct.

Comments on the Quality of English Language

Minor editing of the English language is required.

Round 2

Reviewer 1 Report

Comments and Suggestions for Authors

In my opinion the manuscript has been corrected, improved according to reviewers. 

Reviewer 2 Report

Comments and Suggestions for Authors

Manuscript has been improved and can be published

Comments on the Quality of English Language

 Minor editing of English language required

Reviewer 3 Report

Comments and Suggestions for Authors

All is OK.
